# Rotavirus vaccine impact assessment surveillance in India: protocol and methods

Nayana P Nair,[1] Samarasimha Reddy N,[1] Sidhartha Giri,[1] Venkata Raghava Mohan,[2] Umesh Parashar,[3] Jacqueline Tate,[3] Minesh Pradyuman Shah,[3] Rashmi Arora,[4,5] Mohan Gupte,[4] Sanjay M Mehendale,[4,6] Investigators of the Rotavirus vaccine Impact Surveillance Network,[1] Gagandeep Kang[1,5]

[1]Department of GI Sciences, Christian Medical College, Vellore, India
[2]Department of Community Health, Christian Medical College, Vellore, India
[3]Centers for Disease Control and Prevention, Atlanta, Georgia, USA
[4]Indian Council of Medical Research, New Delhi, India
[5]Translational Health Science and Technology Institute, Faridabad, India
[6]National Institute of Epidemiology, Chennai, India

**Correspondence to**
Professor Gagandeep Kang;
gkang@cmcvellore.ac.in,
gkang@thsti.res.in

## ABSTRACT

**Introduction** Rotavirus infection accounts for 39% of under-five diarrhoeal deaths globally and 22% of these deaths occur in India. Introduction of rotavirus vaccine in a national immunisation programme is considered to be the most effective intervention in preventing severe rotavirus disease. In 2016, India introduced an indigenous rotavirus vaccine (Rotavac) into the Universal Immunisation Programme in a phased manner. This paper describes the protocol for surveillance to monitor the performance of rotavirus vaccine following its introduction into the routine childhood immunisation programme.

**Methods** An active surveillance system was established to identify acute gastroenteritis cases among children less than 5 years of age. For all children enrolled at sentinel sites, case reporting forms are completed and a copy of vaccination record and a stool specimen obtained. The forms and specimens are sent to the referral laboratory for data entry, analysis, testing and storage. Data from sentinel sites in states that have introduced rotavirus vaccine into their routine immunisation schedule will be used to determine rotavirus vaccine impact and effectiveness.

**Ethics and dissemination** The Institutional Review Board of Christian Medical College, Vellore, and all the site institutional ethics committees approved the project. Results will be disseminated in peer-reviewed journals and with stakeholders of the universal immunisation programme in India.

## BACKGROUND

Diarrhoea is the fourth leading cause of global under-five mortality and accounts for 9% of deaths among this age group.[1] Rotavirus infection accounts for 39% of these diarrhoeal deaths, the majority of which occur in low and middle-income countries.[2] India accounts for 22% of the total global rotavirus mortality.[3] As per the Global Burden of Disease Study, 21 357.6 (13 150.8–33 967.0) deaths occur children less than 5 years in India due to rotavirus infection.[4] Rotavirus is a leading cause of moderate to severe acute

### Strengths and limitations of this study

► First project in India to evaluate the impact and effectiveness of a newly introduced vaccine in children.
► Use of test negative control design, a convenient and low-cost technique used in effectiveness studies for several vaccines, including rotavirus vaccines.
► Hospital based surveillance might not be representative of illness or vaccine coverage in the community.

diarrhoea in India; it accounts for 24% of cases of diarrhoea among children less than 23 months of age and 13% of cases of diarrhoea among children aged 24–59 months of age.[5 6] An estimated 11.4 million episodes of rotaviral gastroenteritis occur among under five children leading to 872 000 hospitalisations annually.[7] Introduction of rotavirus vaccine in a national immunisation programme is considered to be the most effective intervention in preventing severe rotavirus disease.[8 9]

The Indian National Rotavirus Surveillance Network, established by the Indian Council of Medical Research (ICMR) in collaboration with US Centers for Disease Control and Prevention (CDC), has played an important role in documenting the disease burden of rotavirus hospitalisations in India. Established in 2005 with seven states conducting rotavirus surveillance, the network expanded over subsequent years.[10] By 2012, 17 states and two union territories in India were conducting rotavirus surveillance using a standardised protocol.[11] The surveillance data highlighted the high prevalence (40%) of rotavirus in children hospitalised with diarrhoea, with representation across the country.[11] Rotavirus was also found to cause significant disease burden in children <5 years of age treated

for diarrhoea as outpatients,[12] with a study in Kolkata reporting 48% positivity for rotavirus over 36 months.[13]

Since 2006, two live attenuated, orally administered rotavirus vaccines have been available globally—a monovalent human rotavirus vaccine [RV1; Rotarix (GSK Biologicals, Rixensart, Belgium)] and a pentavalent bovine-human reassortant vaccine [RV5; RotaTeq (Merck and Co, West Point, PA, USA)].[14 15] The WHO recommended inclusion of these vaccines in the routine immunisation programme of all countries by 2009.[16] Vaccination was considered cost-effective in India but concerns about affordability and long-term price sustainability remained.[17 18] In March 2014, the results of the efficacy and tolerability trial of the first Indian-manufactured oral rotavirus vaccine (Rotavac, Bharat Biotech) were reported.[19] Vaccine efficacy against severe rotavirus gastroenteritis in children up to 2 years of age was 55.1% (95% CI 39.9 to 66.4) for three doses of the vaccine given at 6, 10 and 14 weeks of age.[20] The National Technical Advisory Group on Immunisation recommended the inclusion of Rotavac into the Universal Immunisation Programme (UIP) in 2014 based on the data from surveillance studies and the clinical trial.[10 19 20] Rotavac is available at about 64 rupees per dose which is almost one-tenth the cost of the same vaccine in the private market and less than other rotavirus vaccines.[19] The Ministry of Health and Family Welfare accepted this recommendation, procured vaccine and conducted training for an early phase introduction in the states of Odisha, Andhra Pradesh, Haryana and Himachal Pradesh by April 2016.[21] In 2017, the introduction was extended to five more states, Rajasthan, Madhya Pradesh, Assam, Tripura and Tamil Nadu and will further expand to Uttar Pradesh by 2018.[22] The coverage of the third dose of rotavirus vaccine as per WHO and United Nations Children's Fund (Unicef) estimates for 2017 in India is about 60%.[23]

The impact and effectiveness of rotavirus vaccine in other countries and regions has been studied within surveillance networks by following trends in rotavirus diarrhoea hospitalisations prevaccine and postvaccine introduction and/or by case–control approaches to estimate vaccine effectiveness.[24–27] Furthermore, since rotavirus accounts for a substantial proportion of all-cause and especially severe diarrhoea in children, all-cause diarrhoeal hospitalisation rates can also be used to estimate the number of hospitalisations and outpatient visits prevented by vaccine introduction.

With the introduction of an indigenous vaccine into India's UIP, issues including vaccine impact on disease burden under conditions of routine use, effectiveness against currently circulating strains of rotavirus, safety of the vaccine with respect to intussusception and cost effectiveness of the vaccination programme need to be examined. Studies to examine rotavirus vaccine impact and safety using proven study designs can help answer these questions and provide support for broader introduction of rotavirus vaccine in India.[28] The paper describes the protocol and implementation of an ongoing multisite surveillance to monitor the impact and effectiveness of rotavirus vaccine following its introduction into the routine childhood immunisation programme.

## OBJECTIVES

The primary objectives are:

1. To identify cases of rotavirus among children less than 5 years hospitalised for acute gastroenteritis (AGE) and to determine the circulating rotavirus genotypes preintroduction and postintroduction of Rotavac vaccine using sentinel hospital-based active surveillance.
2. To measure changes in proportion of AGE due to all causes including rotaviral gastroenteritis and severity of presentations at the sentinel surveillance sites preintroduction and postintroduction of Rotavac.
3. To determine the effectiveness of a completed series of Rotavac against laboratory-confirmed severe rotavirus AGE under conditions of routine use in India, using sentinel hospital surveillance sites and case-control methods.

The secondary objectives are:

1. To determine vaccine effectiveness against specific rotavirus genotypes.
2. To determine vaccine effectiveness of a partial series of Rotavac.
3. To determine potential waning of Rotavac effectiveness during the surveillance period.

## METHODS/DESIGN

### Project management

The rotavirus vaccine impact assessment project is funded by the Bill and Melinda Gates Foundation and conducted by the Christian Medical College (CMC), Vellore and the CDC, Atlanta, USA in collaboration with the ICMR, New Delhi and the Translational Health Science and Technology, Institute (THSTI). CMC is responsible for all administrative arrangements, while monitoring is jointly conducted by CMC, ICMR and THSTI. Prior to the vaccine introduction, the Ministry of Health and Family Welfare and the ICMR wrote to the Health Secretaries in each state to inform them of the impact assessment surveillance and request their cooperation.

### Site selection

The processes for sites participating in assessment of vaccine impact and effectiveness include identification and reporting of childhood diarrhoea and laboratory detection of rotavirus from stool samples. When selecting sites for participation, priority was given to sites that had previously conducted surveillance under the National Rota Surveillance Network (NRSN). To identify additional sites, we selected large tertiary care hospitals in states introducing vaccine and requested their participation. A meeting was organised for potential site investigators and site representatives were requested to present retrospective data on diarrhoea admissions among children under 5 years of age (online supplementary table

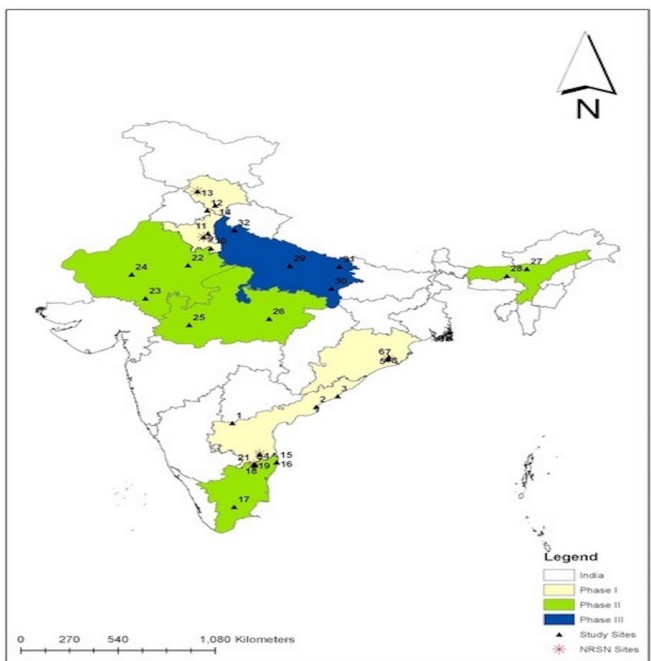

**Figure 1** Location of surveillance sites in states introducing Rotavac into the Universal Immunisation Programme. 1, Kurnool; 2, Kakinada; 3, Vishakhapatnam; 4, Tirupati; 5, Cuttack; 6–8, Bhubaneswar; 9, Rohtak; 10, Mewat; 11, Sonipat; 12, Shimla; 13, Tanda; 14, Chandigarh; 15, 16, Chennai; 17, Madurai; 18–21, Vellore; 22, Jaipur; 23, Udaipur; 24, Jodhpur; 25, Indore; 26, Jabalpur; 27, Tezpur; 28, Guwahati; 29, Lucknow; 30, Varanasi; 31, Gorakhpur; 32, Bijnor. NRSN, National Rota Surveillance Network sites.

1). The criterion for a site to participate in AGE surveillance was at least 250 under five diarrhoea admissions per year. A few sites like Varanasi and Tezpur, with lower numbers were included based on their geographic location and lack of other sites. On average, of seven hospitals contacted in each state, three hospitals met these criteria and were selected (figure 1). For each selected participating hospital, a Memorandum of Understanding was signed between CMC and the participating institution.

Each participating site recruited a four-member team consisting of a project medical officer, a laboratory technician and two field workers. During the initiation visit, meetings were held with paediatrics and community health departments where the project tools, logistics and possible timelines were discussed. The initial contact to initiation of the surveillance took at least 6 months for completing institutional processes and hiring and training of staff (figure 2).

### Ethical clearance
Ethical clearances were obtained from the Institutional Review Board of CMC, Vellore, and from each participating institution (online supplementary table 2).

### Project design and setting
This is a multi-centric surveillance project being carried out in nine states of India over a period of 4 years (figure 1). Vaccine effectiveness will be determined by

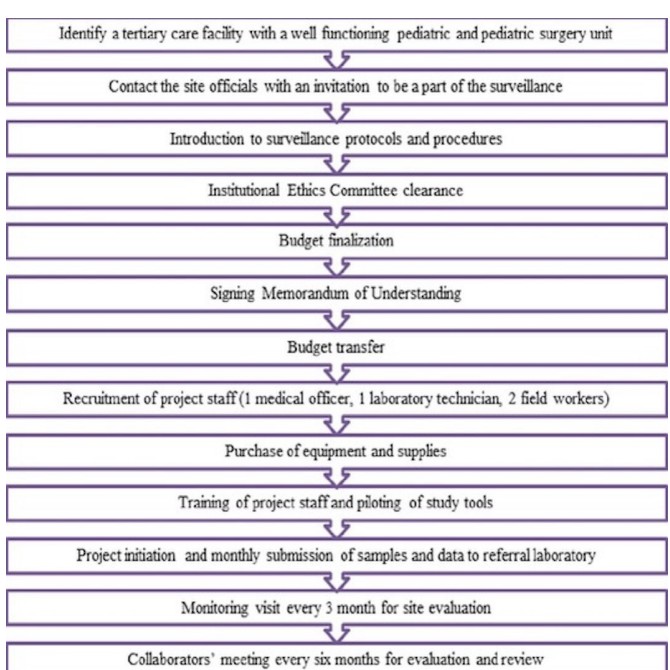

**Figure 2** Process for establishment of a new surveillance site.

a case-control evaluation. Vaccine impact will be determined by monitoring trends in rotavirus hospitalisations prerotavirus and postrotavirus vaccine introduction at the five sites that have preintroduction data (table 1). The other sites will have 4 years of surveillance after vaccine introduction.

### AGE case definition
AGE is defined as the occurrence of ≥three episodes of diarrhoea within a 24 hours period, for less than 7 days prior to the hospital visit, not explained by an underlying medical condition. The definition of 'looser than normal stool' is based on parental perception/judgement of their child's illness and might include changes in consistency and/or volume or changing habits of the child (especially among young/breast fed infants).[29] Severe AGE will be defined as any case with a Vesikari severity score of ≥11.[30]

### Inclusion criteria
Children are eligible for enrolment in the surveillance programme if they meet all of the following criteria: (1) present to one of the active surveillance hospitals for treatment of AGE, (2) are <5 years of age, (3) produce a stool sample during the first 48 hours after the presentation and (4) the child is either treated with oral or intravenous fluids in the emergency department for at least 6 hours or admitted to the hospital and treated with oral rehydration or intravenous fluids.

### Exclusion criteria
Patients will be excluded from enrolment if any of the following apply: (1) more than 5 years of age, (2) unable to obtain informed consent from the parent/caregiver or guardian, (3) admitted to another hospital for >24 hours

**Table 1** Participating sites in the rotavirus vaccine impact assessment surveillance

| S. No. | Surveillance Network Centers | District | State | Date of vaccine introduction | Phase |
|---|---|---|---|---|---|
| 1 | Kurnool Medical College | Kurnool | Andhra Pradesh | 20 April 2016 | Phase I |
| 2 | Government General Hospital | Kakinada | Andhra Pradesh | 20 April 2016 | Phase I |
| 3 | Andhra Medical College | Vishakhapatnam | Andhra Pradesh | 20 April 2016 | Phase I |
| 4 | Sri Venkateswara Medical College* | Tirupati | Andhra Pradesh | 20 April 2016 | Phase I |
| 5 | Sardar Vallabhbhai Patel Post Graduate Institute of Paediatrics | Cuttack | Odisha | 26 March 2016 | Phase I |
| 6 | Kalinga Institute of Medical Sciences | Bhubaneswar | Odisha | 26 March 2016 | Phase I |
| 7 | Institute of Medical Sciences and SUM Hospital | Bhubaneswar | Odisha | 26 March 2016 | Phase I |
| 8 | Hi-Tech Hospital and Medical College* | Bhubaneswar | Odisha | 26 March 2016 | Phase I |
| 9 | Pandit Bhagwat Dayal Sharma Post Graduate Institute of Medical Sciences* | Rohtak | Haryana | 11 April 2016 | Phase I |
| 10 | Shaheed Hasan Khan Mewati Government Medical College | Mewat | Haryana | 11 April 2016 | Phase I |
| 11 | BPS Government Medical College for Women | Sonipat | Haryana | 11 April 2016 | Phase I |
| 12 | Post Graduate Institute of Medical Education and Research | Chandigarh | Chandigarh | 15 March 2016 | Phase I |
| 13 | Rajendra Prasad Government Medical College* | Tanda | Himachal Pradesh | 05 December 2015 | Phase I |
| 14 | Indira Gandhi Government Medical College | Shimla | Himachal Pradesh | 11 April 2016 | Phase I |
| 15 | Kanchi Kama Koti Child Trust Hospital | Chennai | Tamil Nadu | 20 September 2017 | Phase II |
| 16 | Institute of Child Health | Chennai | Tamil Nadu | 20 September 2017 | Phase II |
| 17 | Government Medical College | Madurai | Tamil Nadu | 20 September 2017 | Phase II |
| 18 | Christian Medical College* | Vellore | Tamil Nadu | 20 September 2017 | Phase II |
| 19 | Government Vellore Medical College | Vellore | Tamil Nadu | 20 September 2017 | Phase II |
| 20 | Nalam Hospital | Vellore | Tamil Nadu | 20September 2017 | Phase II |
| 21 | Narayani Hospital and Research Centre | Vellore | Tamil Nadu | 20 September 2017 | Phase II |
| 22 | Sawai Man Singh Medical College | Jaipur | Rajasthan | 23 March 2017 | Phase II |
| 23 | Rabindranath Tagore Medical College | Udaipur | Rajasthan | 23 March 2017 | Phase II |
| 24 | Dr. Sampurnanand Medical College | Jodhpur | Rajasthan | 23 March2017 | Phase II |
| 25 | Mahatma Gandhi Memorial Medical College | Indore | Madhya Pradesh | 02 April 2017 | Phase II |
| 26 | Netaji Subhash Chandra Bose Medical College | Jabalpur | Madhya Pradesh | 02 April 2017 | Phase II |
| 27 | Baptist Christian Hospital | Tezpur | Assam | 14 June 2017 | Phase II |
| 28 | Government Medical College | Guwahati | Assam | 14 June 2017 | Phase II |
| 29 | King George Medical College | Lucknow | Uttar Pradesh | 16 July 2018 | Phase III |
| 30 | Institute of Medical Sciences, Banaras Hindu University | Varanasi | Uttar Pradesh | 16 July 2018 | Phase III |
| 31 | BRD Medical College | Gorakhpur | Uttar Pradesh | 16 July 2018 | Phase III |
| 32 | Mangala Hospital and Research Centre | Bijnor | Uttar Pradesh | 16 July 2018 | Phase III |

*Sites that were part of the National Rota Surveillance Network project.

(and subsequently transferred to the current sentinel site), (4) children from states without Rotavac in their immunisation schedule (immigrant population).

### Vaccine effectiveness case-control evaluation

The vaccine effectiveness assessment against rotavirus hospitalisations will be conducted through a test-negative case-control design.[31] The test negative study design is a convenient and low cost method for estimation of effectiveness of a vaccine as the study can be facility-based rather than community-based and since commercially available ELISA kits have a sensitivity and specificity of over 99%,[32] the probability of any bias is negligible.[33] For the purpose of the case-control study, a child is considered vaccinated if he/she has received at least one dose of vaccine at least 14 days before gastroenteritis onset or hospital admission.[19]

Cases and controls are eligible for inclusion in the vaccine effectiveness case-control evaluation if they are: (1) enrolled in the active surveillance platform of a sentinel hospital, (2) age eligible to have received the

vaccine, that is, at least 6 weeks of age and born after the date of vaccine introduction. Hence, children aged less than 42 days at the time of vaccine introduction will be included in the vaccine effectiveness evaluation. Cases will have a stool specimen that tests positive for rotavirus by ELISA and controls will have a stool specimen that tests negative for rotavirus by ELISA. Cases and controls will be excluded from the vaccine effectiveness evaluation if they are enrolled prior to vaccine introduction or are age ineligible to have received vaccine.

With regard to vaccination, efforts will be made to obtain a vaccination card, medical record or other form of documentation through which rotavirus vaccine status may be verified. Children without a vaccination record will be excluded from the vaccine effectiveness analysis.

## Sample size

As trends are being monitored, continuous enrolment of children presenting to the sentinel hospitals during the evaluation period will be done. Based on recent figures, the sentinel hospitals should admit annually a combined 600–1000 children less than 5 years of age with AGE per state, of which approximately 200–300 children are expected to have severe rotavirus gastroenteritis.

For the vaccine effectiveness evaluation, the sample size is calculated to achieve 80% power at the 5% significance level to detect OR of 0.6 (vaccine effectiveness of 40%). The ratio of cases to controls will be 1:2. Expecting the vaccine coverage to be 80% or above for a full series of vaccinations, we would need approximately 242 cases and 484 controls to demonstrate a vaccine effectiveness of ≥40%.[23 34] To enable further analyses such as genotype-specific vaccine effectiveness calculations, case recruitment will continue throughout the duration of the project irrespective of numbers achieved.

## Surveillance activities

Per the surveillance protocol, project staff visit hospital wards daily to enrol cases admitted with acute diarrhoea. Details of all inpatient diarrhoea cases (0–59 months) are recorded in the diarrhoea logbook. For all cases fulfilling eligibility criteria, informed consent is taken from parents/caregivers. Sociodemographic, clinical and vaccination data are collected, along with a stool sample from the child. The interviewer reviews and photographs or photocopies the vaccination record and documents the date of administration of rotavirus vaccine, as well as other vaccines received. If the vaccination card is not available for enrolled children at the time of interview, site staff visits the child's home to review the record. If the vaccination card is not available, staff seeks more information through a visit to the local health unit where vaccinations were administered. Depending on the site, different methods including smartphones, emails, or self-addressed envelopes with stamp and money are also employed to ensure recording of accurate information. We are currently able to procure 85% of vaccination cards from enrolled children.

## Data flow

Stool samples and completed case report forms are sent to the referral laboratory at CMC, Vellore, monthly for testing, data entry and storage. On receipt of samples and case report forms, the shipment is checked and logged, and any issues with quantity or quality recorded. Case report forms are reviewed before data entry and data clarification forms are generated in case of any missing fields or errors and sent to sites within 3 days of receipt for clarification and responses. At the referral laboratory, stool samples are tested for rotavirus antigen using commercially available ELISA kits (Rotaclone; Meridian Biosciences, USA) and genotyped using published methods.[35–37] Rotavirus test results are reported to the sites within 1 month of receiving the samples and forms.

## Monitoring surveillance

For the first 3 months after initiation of surveillance, sites are visited monthly. After 3 months, monitoring is reduced to once in every 3 months. At each visit, site evaluation is conducted using a monitoring checklist (online supplementary table 3), which records performance in terms of enrolment of children with diarrhoea, collection of adequate stool samples and obtaining a copy of the vaccination card from all enrolled children. Every 6 months, a collaborators' meeting is organised to discuss the work done by each site and to enable collaborators to provide feedback to individual sites.

## Analysis plan
### AGE surveillance

Descriptive analyses of demographic, clinical and treatment information will be performed to describe the children enrolled in the active surveillance programme and children with specimens testing positive for rotavirus. Trends in all-cause diarrhoea will be compared for prevaccine and postvaccine introduction using available population demographics for five NRSN sites, Tirupati (SVMC); Bhubaneswar (Hitech hospital); Rohtak (PGIMS); Tanda (RPGMC) and Vellore (CMC) that have hospital-based pre-vaccine data on AGE among under five children from 2012 onwards. For these five sites, a simple comparison of rotavirus diarrhoea before and after the vaccine introduction will be estimated. The distributions of circulating genotypes will be described over the period and compared prevaccine and postvaccine introduction. Data will be stratified by age eligibility to receive the vaccine for examining possible indirect effects among older children.

### Vaccine effectiveness case-control evaluation

The primary analysis will include all verified reports of vaccine status for children who have received a full vaccine series versus no doses of the vaccine. An unconditional logistic regression controlling for the date of birth and age in the model will be used. Vaccine effectiveness will be estimated using the formula $[(1-OR) \times 100\%]$, where the OR is the adjusted OR for the rotavirus immunisation rate among case-patients compared with control.[38 39] If

sample size permits, genotype-specific vaccine effectiveness will be calculated for the predominant circulating strains detected after vaccine introduction.

## PATIENT and public involvement

Patients and the public were not involved in the design of the study, which is being conducted under the supervision of the government to assess vaccine impact. Data will be shared as press release and publications when available.

## ETHICS AND DISSEMINATION

All parents/legal guardians whose children have been hospitalised for AGE receive both oral and written information about the project. Parents/legal guardians willing to participate provide informed consent for their child's participation. Participants receive unique IDs which are used for questionnaires. The record connecting IDs with names is securely stored separate from the data, with access only for authorised personnel. All other investigators access the password protected data set without personal identifiers.

The results from the project will be communicated with all the national stakeholders, viz, Ministry of Health and Family Welfare, Government of India, the National Technical Advisory Group of Immunizations and the State Health Secretaries. Results will be disseminated in international and national peer-reviewed journals. The results will also be communicated at meetings and conferences.

## DISCUSSION

### Challenges in establishing sites for impact assessment

Sites that had not previously participated in NRSN were sometimes reluctant to participate, particularly those located in remote areas. Recruitment of project staff was difficult, because of lack of suitably qualified potential staff. Staff turnover rate was high in certain hospitals, especially in rural areas. Some governmental institutions had difficulty in being able to use project funds, because their administration lacked experience in handling research projects. Recruitment and retention of project staff, particularly medical officers, is an ongoing challenge.

### Challenges in project implementation

Lack of experience among site investigators in prior research resulted in the need to consistently monitor data quality. Communication with sites, especially in hilly and rural sites, is difficult, particularly with regard to telephone and email communication because of poor connectivity. These issues can be partially addressed through the use of mobile hot spots. Transport of biological specimens is an issue with courier companies, particularly because samples are shipped in temperature-controlled containers.

Collection of immunisation cards is crucial for the impact assessment and these cards are often not easily obtained. Therefore, multiple measures had to be implemented to ensure collection of the cards. When rotavirus vaccination was introduced, older vaccination cards available within the vaccination programme did not have space for recording the new vaccine. This required careful enquiry and verification of rotavirus vaccine receipt with the registers of Anganwadi workers,[40] who are responsible for maintaining governmental immunisation records for rural children.

Initially, the Rotavac vaccine was issued as ten dose vials. Since there is no open phial policy for the vaccine, the peripheral centres withheld vaccination until they had a minimum of seven infants before opening a new phial. In the initial period of the project, coverage was therefore low in the target age group. This was subsequently addressed by issuing five dose vials, and recommending immunisation irrespective of the number of children to be immunised.

In data collection, common issues encountered include missing fields on case reporting forms, incorrect labelling of samples, inaccurate screening of hospital logs, which resulted in missing potential cases for enrolment, and errors in recording the immunisation history of enrolled children. Frequent monitoring and retraining during the initial year of surveillance resulted in an improvement in performance across sites.

A limitation of this project is that rotavirus vaccination with other commercially available oral rotavirus vaccines started in the private sector, and some coverage may have been achieved before the indigenous vaccine's introduction into the country's UIP. Herd protection has been seen in other settings,[41] and if high coverage has been achieved with other rotavirus vaccines, it may be difficult to demonstrate effectiveness of Rotavac. It is also possible that low vaccination coverage in the public immunisation system may result in a need to modify the analytic plan. Further, the use of test-negative controls as compared with community controls may result in minimal bias if vaccination coverage is different among those who access the sentinel healthcare facilities and those who do not.[33]

## CONCLUSIONS

This description of the protocol and challenges to extend a surveillance platform to monitor the impact and effectiveness of a newly introduced vaccine highlights the importance of robust surveillance systems for vaccine preventable diseases. Such a platform is critical to document and study the performance of a new vaccine introduced into the immunisation programme. The rotavirus vaccine impact surveillance is expected to generate data regarding changes in proportion of AGE due to all causes including rotaviral gastroenteritis and severity of presentations at the sentinel surveillance sites before and after the introduction of Rotavac, as well as permit a case-control analysis, using a test negative design. The data that we will obtain from different sites and states will help in measuring the effectiveness of the vaccine in routine programmatic use, but conduct of high-quality impact

assessment requires attention to process with a need to identify and apply site-specific solutions.

**Acknowledgements** The authors would like to thank the investigators of all the participating institutions and the children and their families who are a part of the project. They thank Mr J Senthil Kumar for his help with the maps. The investigators of rotavirus vaccine impact assessment surveillance network are Sowmiya VS, Rama Prasad GS, Goru Krishna Babu, Padmalatha Pamu, B Manohar, Subal Pradhan, Mrutunjay Dash, Nirmal Kumar Mohakud, Rajib Ray, Geetha Gathwala, Suraj Chawla, Manoj Rawal, Madhu Gupta, Sanjeev S Choudhary, Shyam Kaushik, S Balasubramaniyan, CP Girish Kumar, Sridevi A Narayan, Kulaindaivel S, Anna Simon, RK Gupta, Suresh Goyal, Pramod Sharma, Sharad Thora, Pawan Ghanghoriya, Koshy George, Jayanta Goswami, Ashish Wakhlu, Vineeta Gupta, Mahima Mithal and Vipin Vashishtha.

**Collaborators** The investigators of Rotavirus vaccine impact assessment surveillance network are Sowmiya VS, Rama Prasad GS, Goru Krishna Babu, Padmalatha Pamu, B Manohar, Subal Pradhan, Mrutunjay Dash, Nirmal Kumar Mohakud, Rajib Ray, Geetha Gathwala, Suraj Chawla, Manoj Rawal, Madhu Gupta, Sanjeev S Choudhary, Shyam Kaushik, S Balasubramaniyan, CP Girish Kumar, Sridevi A Narayan, Kulaindaivel S, Anna Simon, R K Gupta, Suresh Goyal, Pramod Sharma, Sharad Thora, Pawan Ghanghoriya, Koshy George, Jayanta Goswami, Ashish Wakhlu, Vineeta Gupta, Mahima Mithal and Vipin Vashishtha.

**Contributors** NPN, SNR, JT, MPS, UP, MG, SMM and GK conceived this paper. VRM, GK, RA, SG, NPN and SNIRvi obtained all the approvals for the project. SG and GK developed and approved all the laboratory protocols. NPN wrote the first draft. All authors read, critically revised and approved the final manuscript. The Investigators of the Rotavirus vaccine Impact Surveillance Network are responsible for study activities and data collection from respective sentinel sites.

**Funding** This work was supported by grants from the Bill and Melinda Gates Foundation to the Centers for Disease Control and Prevention, Atlanta, GA, USA (subcontract to CMC Vellore grant no MOA#871-15SC) and the Translational Health Science and Technology Institute (grant no OPP1165083).

**Disclaimer** The findings and conclusions in this report are those of the authors and do not necessarily represent the official position of the US Centers for Disease Control and Prevention.

The depiction of boundaries on the map(s) in this article do not imply the expression of any opinion whatsoever on the part of BMJ (or any member of its group) concerning the legal status of any country, territory, jurisdiction or area or of its authorities. The map(s) are provided without any warranty of any kind, either express or implied.

**Competing interests** None declared.

**Ethics approval** The project was approved by the ethics committee of CMC, Vellore and also by the ethical committees of all the participating institutes.

**Provenance and peer review** Not commissioned; externally peer reviewed.

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
