## [Reviewer comments · BMJ Open]

ARTICLE DETAILS

TITLE (PROVISIONAL)	Rotavirus vaccine impact assessment surveillance in India: Protocol and methods
AUTHORS	Nair, Nayana; Reddy, Samarasimha; Giri, Sidhartha; Mohan, Venkata Raghava; Parashar, Umesh; Tate, Jacqueline; Shah, Minesh; Arora, Rashmi; Gupte, Mohan; Mehendale, Sanjay; Investigators of the Rotavirus vaccine impact, Surveillance Network; Kang, Gagandeep

VERSION 1 - REVIEW

REVIEWER	Dr Daniel Hungerford Research Fellow Institute of Infection and Global Health, University of Liverpool, UK
REVIEW RETURNED	03-Jul-2018

GENERAL COMMENTS	This is a well written piece by Nair et al., which provides an easy to follow protocol for rotavirus vaccine impact evaluation via surveillance. It is clearly essential to conduct a thorough evaluation of a new vaccine programme particularly given the baseline burden of rotavirus disease in India, therefore this an important study. In line with BMJ Open Protocol publishing guidance the manuscript contains no data or conclusions. I do have some comments / questions below which I feel will enhance the paper and need clarification. As per protocol guidance please could the authors add some timelines for the study and the data, how much pre- and post-data will be available for each site? The authors mention that if there is low or high vaccine coverage (Pg 16 line 52 and Pg 17 line 3-4) there may need to be a change to analytical plan. Presumably in this scenario this would add limitations to the case-control study because of any of the following, for example; herd effect (lack of cases), different exposure risk between vaccinated and unvaccinated or larger sample size needed. In reference to this, can the authors explain why surveillance was not/is not being conducted in areas without vaccination as these states could potentially be used as natural population level geographical controls? Is it because there is state based heterogeneity in the baseline rate of hospitalised rotavirus disease?
---

	Background Pg 5 line 3 and line 27. Figures for how much diarrhea rotavirus accounts for are very different, 24% and 13% vs 40% please provide justification or comment on the difference Pg 5 line 32 and 34. Please add what these patients were outpatients for e.g. diarrhea. Pg 5 line 53. Please make it clear that Rotavac is also an oral vaccine. Methods Table 1: be useful for the reader if the date of vaccine introduction could be added for each state. Pg 6 lines 15-24. Can the authors add what the vaccine coverage is to date as this would be useful for assessing the feasibility of the case control study and assumptions for the sample size calculation. Pg 12 line 43. See above. Please justify the “expecting vaccine coverage to be 80%...” with a reference. Analysis Plan Pg14 line 30. Please add how you intend to compare pre and post vaccine rates. Discussion Pg 16 Line 36. The authors mention commercially available Rotarix and RotaTeq, are the authors able to quantify the coverage? Pg 17 line 10. As I recall VE estimates against hospitalised rotavirus disease estimated using test-negative or community controls were very similar, in both Malawi and the US, for example. Can the author’s reference systematic reviews which have included comparisons in effectiveness between community and test negative hospital controls? It may strengthen the rationale for using test-negative controls in this study.
--	---

REVIEWER	thomas snelling TKI, Australia
REVIEW RETURNED	25-Jul-2018

GENERAL COMMENTS	This is clearly written and mostly straight-forward. There is a paragraph on ethics and dissemination between the discussion and conclusion which I would have thought should come before the discussion. I have provided some minor comments directly on the attached manuscript which should be fixed or clarified. The reviewer provided a marked copy with additional comments. Please contact the publisher for full details.
--

REVIEWER	Prof Kristine Macartney National Centre For Immunisation Research & Surveillance
-----------------	---

GENERAL COMMENTS

Overall very well constructed protocol and well written manuscript. This protocol describes how post Rotavac (new rotavirus) vaccine introduction will be assessed with respect to impact on all-cause AGE, circulating rotavirus genotypes, and vaccine effectiveness. Some minor points of note:

- Is it possible to give an approximate estimate of the number of children affected by rotavirus in India. The first two paragraphs contain a range of %, but an absolute estimate (range) would be good to include
- bottom page 5 and later in manuscript authors allude to need to monitor safety, especially impact on intussusception rates. I realise this is not the topic of this study, but mention in Discussion on plan/s to monitor impact on IS would be of interest
- pg 7 states all sentinel sites selected had at least 250 diarrhea admissions annually, but Table shows sites with lower or no numbers
- pg 10, should the inclusion criteria include min age of 6 weeks also? Criteria for inclusion in the VE study implies a minimum age of 6 weeks at vaccination + 14 days
- could the authors discuss what their experience/success rate is in obtaining a verified vaccination record? (pg 11/12)
- pg 12 - could authors confirm if stool samples are only refrigerated, not frozen? As they are only shipped on a monthly basis for testing, could viral degradation at refrigerated (as compared with freezer) temperatures lead to false negative results?
- pg 11 - does the power calculation for the VE study include loss of participants due to vaccination status not being able to be identified? Similarly, has the proportion not having a stool test result been accounted for? Also, does it factor in lower case numbers occurring in the later years of the study (assuming decreases in RV-AGE)
- line 12 - does daily = 7 days per week? or weekdays?
- pg 13 lines 34-37 - sentence not clear
- in the VE analysis, could authors describe if and how they will pool data across states? How will you control for the differing impacts of herd immunity in states where the timing of post vaccination assessment varies between states.
- pg 15, line 8 When was Rotavac first introduced? Also, is there an upper age limit/s for administration?
- pg 15, unclear why there would be a 'delay' in recruitment of cases due to use of non Rotavac vaccines in private market? Could authors comment more on this, Is there any estimate of private market vaccine utilisation in these settings?
- Suppl Table 2 Project activities table1. Meaning of row 2 "source from which..." is unclear. Does this mean staff member? Also under table2, what aspect of form does 'legibility' refer to and Table 5. How is storage of samples assessed.
- In summary indicators, as per above, can the study team reference past experience in getting >90% stool collection and >75% vaccination records.

VERSION 1 – AUTHOR RESPONSE

Reviewer(s)' Comments to Author:

Reviewer: 1

Reviewer Name: Dr Daniel Hungerford

Institution and Country: Institute of Infection and Global Health, University of Liverpool, UK

Please state any competing interests or state 'None declared': None declared

Please leave your comments for the authors below

This is a well written piece by Nair et al., which provides an easy to follow protocol for rotavirus vaccine impact evaluation via surveillance. It is clearly essential to conduct a thorough evaluation of a new vaccine programme particularly given the baseline burden of rotavirus disease in India, therefore this an important study.

In line with BMJ Open Protocol publishing guidance the manuscript contains no data or conclusions.

I do have some comments / questions below which I feel will enhance the paper and need clarification.

Reviewers comment: 1. As per protocol guidance please could the authors add some timelines for the study and the data, how much pre- and post-data will be available for each site?

Response: Data collection will be for 4 years from each site. Five sites in the study (PGIMS, Rohtak; RPGMC, Tanda; Hitech hospital, BBSR; SVMC, Tirupati and CMC, Vellore) have 3-5 years data of prior to vaccine introduction. All the other sites have data after the introduction of vaccine.

The text now reads "Vaccine impact will be determined by monitoring trends in rotavirus hospitalizations pre- and post-rotavirus vaccine introduction at the five sites that have pre-introduction data [Table.1]. The other sites will have four years of surveillance after vaccine introduction." In Pg 8 lines 13-14.

Reviewers comment: 2. The authors mention that if there is low or high vaccine coverage (Pg 16 line 52 and Pg 17 line 3-4) there may need to be a change to analytical plan. Presumably in this scenario this would add limitations to the case-control study because of any of the following, for example; herd effect (lack of cases), different exposure risk between vaccinated and unvaccinated or larger sample size needed. In reference to this, can the authors explain why surveillance was not/is not being conducted in areas without vaccination as these states could potentially be used as natural population level geographical controls? Is it because there is state based heterogeneity in the baseline rate of hospitalised rotavirus disease?

In this study, surveillance is not being conducted in areas without vaccination because of state- and site-based heterogeneity in the baseline rate of hospitalized rotavirus disease. This was seen in earlier surveillance and is also being seen in the data being generated in this study, in the post-introduction data from different states and at sites where there is more than one site per state.

Ref: National Rotavirus Surveillance Network, Kumar CPG, Venkatasubramanian S, et al. Profile and Trends of Rotavirus Gastroenteritis in Under 5 children in India, 2012 - 2014, Preliminary Report of the Indian National Rotavirus Surveillance Network. Indian Pediatr 2016;53:619–22.

Background

Reviewers comment: 3. Pg 5 line 3 and line 27. Figures for how much diarrhea rotavirus accounts for are very different, 24% and 13% vs 40% please provide justification or comment on the difference

Line 20, 21 Pg 3 in the manuscript cites data on rotaviral diarrhea among different age groups in India from the GEMS study. Line 10 Pg 4 reports data from the Indian National Rota Surveillance Network (NRSN) for 2012-2014 from 28 sites with an overall rotavirus positivity of 40% among under five children. It may be noted that the two studies had different inclusion criteria.

Reviewers comment: 4. Pg 5 line 32 and 34. Please add what these patients were outpatients for e.g. diarrhea.

We have revised the statement which now reads "Rotavirus was also found to cause significant disease burden in children <5 years of age treated for diarrhea as outpatients [12], with a study in Kolkata reporting 48% positivity for rotavirus over 36 months" Page 4 lines 12, 13.

Reviewers comment: 5. Pg 5 line 53. Please make it clear that Rotavac is also an oral vaccine.

Changed as recommended, page 4, line 21.

Methods

Reviewers comment: 6. Table 1: be useful for the reader if the date of vaccine introduction could be added for each state.

We have incorporated the date of vaccine introduction in Table 1, page 8-9.

Reviewers comment: 7. Pg 6 lines 15-24. Can the authors add what the vaccine coverage is to date as this would be useful for assessing the feasibility of the case control study and assumptions for the sample size calculation.

Coverage estimates by WHO and UNICEF in India for 2017 are about 60% coverage for the third dose of rotavirus vaccine. Page 5, lines 9-11 reads "The coverage of the third dose of rotavirus vaccine as per WHO and United Nations Children's Fund (UNICEF) estimates for 2017 in India is about 60% (23)."

Ref no. 23 http://www.who.int/immunization/monitoring_surveillance/data/ind.pdf

Reviewers comment: 8. Pg 12 line 43. See above. Please justify the "expecting vaccine coverage to be 80%..." with a reference.

The coverage estimates by WHO in India for 2017 for the first dose of OPV/ Pentavalent vaccine is more than 80%, and therefore, we expected similar coverage for oral rotavirus vaccine which is given along with the first dose of OPV/pentavalent. While oral rotavirus coverage has actually been reported as being lower, this is in the early stages of introduction, and we expect coverage rates to rise during the course of the study.

Ref no:23 http://www.who.int/immunization/monitoring_surveillance/data/ind.pdf

Ref no:32 <http://pib.nic.in/newsite/PrintRelease.aspx?relid=181816>

Analysis Plan

Reviewers comment: 9. Pg14 line 30. Please add how you intend to compare pre and post vaccine rates.

Response: We intend to compare the rates of rotavirus positive gastroenteritis among all children enrolled in to surveillance with a collected stool sample. This comparison is for the five sentinel

hospitals namely, Vellore, Rohtak, Tanda, Bhubaneswar and Tirupati where the pre-vaccine data is available from 2012 onwards.

Corresponding changes are made in Pg 13 lines 21, 22 & Pg 14 line 1 which now read: “for five NRSN sites namely, SVMC, Tirupati; Hitech Hospital, Bhubaneswar; PGIMS, Rohtak; RPGMC, Tanda and CMC, Vellore that have pre-vaccine data available from 2012 onwards.”

Discussion

Reviewers comment: 10. Pg 16 Line 36. The authors mention commercially available Rotarix and RotaTeq, are the authors able to quantify the coverage?

Response: As per the collected data until August 2018, the coverage for commercially available rotavirus vaccines (RotaRix and RotaTeq) is less than 1% among all children enrolled into surveillance. The population coverage for these commercially available vaccines is unknown.

Reviewers comment: 11. Pg 17 line 10. As I recall VE estimates against hospitalized rotavirus disease estimated using test-negative or community controls were very similar, in both Malawi and the US, for example. Can the author’s reference systematic reviews which have included comparisons in effectiveness between community and test negative hospital controls? It may strengthen the rationale for using test-negative controls in this study.

An appropriate reference has been added on page 17 line 7 and is provided below.

Ref no:40 Haber M, Lopman BA, Tate JE, et al. A comparison of the test-negative and traditional case-control study designs with respect to the bias of estimates of rotavirus vaccine effectiveness. *Vaccine* 2018;36:5071–6. doi:10.1016/j.vaccine.2018.06.072

Reviewer: 2

Reviewer Name: Thomas Snelling

Institution and Country: TKI, Australia

Please state any competing interests or state ‘None declared’: None declared

Please leave your comments for the authors below

This is clearly written and mostly straight-forward. There is a paragraph on ethics and dissemination between the discussion and conclusion which I would have thought should come before the discussion. I have provided some minor comments directly on the attached manuscript which should be fixed or clarified.

Reviewers comment: due to rotavirus?

Page 6, line 9 now reads “To measure changes in proportion of AGE due to all causes including rotaviral gastroenteritis.....”

Reviewers comment: and?

“and” added, Page 11 line 5.

Reviewers comment: odds ratio?

“odds ratio” included, Page 11 line 22 of revised manuscript.

Reviewers comment: It seems odd to include this after the discussion.

The ethics section has been moved to before the discussion on page 14 lines 17-21 and page 15 lines 1-9.

Reviewer's comment: actually the effect is the opposite. Using test negative controls mitigates against this bias if both cases and test-negative controls are drawn from sentinel sites. Drawing controls from the community could introduce a bias.

An appropriate reference is added for the same for clarification, Pg 17 line 7 of revised manuscript.

Reviewers comment: due to rotavirus?

Page 17, Line 14 now reads 'The rotavirus vaccine impact surveillance is expected to generate data regarding changes in proportion of AGE due to all causes including rotaviral gastroenteritis.....'.

Reviewers comment: Ref 29 needs author

Authors names added, Pg22 lines 9-12.

Reviewer: 3

Reviewer Name: Prof Kristine Macartney

Institution and Country: National Centre For Immunisation Research & Surveillance

Please state any competing interests or state 'None declared': None declared

Please leave your comments for the authors below

Overall very well constructed protocol and well written manuscript. This protocol describes how post Rotavac (new rotavirus) vaccine introduction will be assessed with respect to impact on all-cause AGE, circulating rotavirus genotypes, and vaccine effectiveness.

Some minor points of note:

Reviewers comment: - Is it possible to give an approximate estimate of the number of children affected by rotavirus in India. The first two paragraphs contain a range of %, but an absolute estimate (range) would be good to include

Page 3, lines 17-19 now read "As per the Global Burden of Disease Study, 21357.6 (13150.8-33967.0) deaths occur among children less than five years in India due to rotavirus infection [4]."

Page 3 line 21 & Page 4 lines 1, 2 read as "An estimated 11.4 million episodes of rotaviral gastroenteritis occur among under five children leading to 872,000 hospitalizations annually [7]."

Reviewers comment: - bottom page 5 and later in manuscript authors allude to need to monitor safety, especially impact on intussusception rates. I realise this is not the topic of this study, but mention in Discussion on plan/s to monitor impact on IS would be of interest

Response: We thank the reviewer for suggesting the addition of safety monitoring. The surveillance for safety is on-going. The protocol has been published and is cited on Page 6 line 1.

Reviewers comment: - pg 7 states all sentinel sites selected had at least 250 diarrhea admissions annually, but Table shows sites with lower or no numbers

“A few sites like Varanasi and Tezpur, with lower numbers were included based on their geographic location and lack of other sites “in Pg 7 lines 18, 19.

Reviewers comment: - pg 10, should the inclusion criteria include min age of 6 weeks also? Criteria for inclusion in the VE study implies a minimum age of 6 weeks at vaccination + 14 days

The study enrolls all children from birth till 5 years of age to examine proportion of hospitalizations due to rotavirus gastroenteritis. However, for estimating the vaccine effectiveness, only age eligible children i.e .born after vaccine introduction and/or aged less than 42 days at the time of vaccine introduction, are included.

This now reads as “Hence, children aged less than 42 days at the time of vaccine introduction will be included in the vaccine effectiveness evaluation” in Pg 11 lines 6,7.

Reviewers comment: - could the authors discuss what their experience/success rate is in obtaining a verified vaccination record? (pg 11/12)

Response: We are currently able to trace the vaccination information for 85% of the enrolled children and obtain verified vaccination information. This now reads as “We are currently able to procure 85% of vaccination cards from enrolled children” in Pg 12 lines 18,19.

Reviewers comment: - pg 12 - could authors confirm if stool samples are only refrigerated, not frozen? As they are only shipped on a monthly basis for testing, could viral degradation at refrigerated (as compared with freezer) temperatures lead to false negative results?

Response: Stool samples are stored at -20°C at all the sentinel sites. The referral testing laboratory receives the samples each month and stores the samples at -70°C until testing, which is usually within one to two weeks. Since rotavirus is a double-stranded virus, viral degradation is not a major concern. Published studies have demonstrated RNA recovery after four weeks at room temperature.

Reviewers comment: - pg 11 - does the power calculation for the VE study include loss of participants due to vaccination status not being able to be identified? Similarly, has the proportion not having a stool test result been accounted for? Also, does it factor in lower case numbers occurring in the later years of the study (assuming decreases in RV-AGE)

Response: The sample size for estimating VE was calculated based on the expected vaccine coverages and expected vaccine effectiveness. The power calculation for sample size of VE estimates does not include the loss of participants due to vaccination status not being identified/ stool sample not collected, because this is prospective surveillance and will continue at least till the required number of cases and controls is reached.

Reviewers comment: - line 12 - does daily = 7 days per week? or weekdays?

The surveillance is in place 7 days a week.

- pg 13 lines 34-37 - sentence not clear

The sentence has been revised to read “Trends in all-cause diarrhea and rotavirus diarrhea will be compared for pre- and post-vaccine introduction using available population demographics for five NRSN sites namely, SVMC, Tirupati; Hitech hospital, Bhubaneswar; PGIMS, Rohtak; RPGMC, Tanda and CMC, Vellore that have pre-vaccine data available from 2012” in Pg 13 lines 20-22 and Pg 14 line 1.

Reviewers comment: - in the VE analysis, could authors describe if and how they will pool data across states? How will you control for the differing impacts of herd immunity in states where the timing of post vaccination assessment varies between states.

The data collected from all the states uses the same collection tools to facilitate pooling data. For vaccine effectiveness analysis, only age eligible children i.e .born after vaccine introduction and/or aged less than 42 days at the time of vaccine introduction will be included. Given the 55% efficacy of the vaccine in the phase 3 study, the introduction of the vaccine into new birth cohorts only, it is unlikely that any herd effect will be demonstrable in the early stages of vaccine introduction. However, we will be able to track this by examining data from children less than 5 years of age who are too old to receive the vaccine when introduced.

Reviewers comment: pg 15, line 8 When was Rotavac first introduced? Also, is there an upper age limit/s for administration?

Rotavac was first introduced in Kangra, Himachal Pradesh on Dec 5, 2015 and later expanded to the states of Andhra Pradesh, Odisha, Haryana and Himachal Pradesh by April 2016. The vaccine is administered along with oral polio vaccine and pentavalent vaccine at 6, 10 and 14 weeks of age. The upper age limit is 1 year for the first dose of the vaccine.

Reviewers comment: pg 15, unclear why there would be a 'delay' in recruitment of cases due to use of non Rotavac vaccines in private market? Could authors comment more on this, Is there any estimate of private market vaccine utilisation in these settings?

Response: The sentence has been edited and now reads “A limitation of this project is that rotavirus vaccination with other commercially available oral rotavirus vaccines started in the private sector, and some coverage may have been achieved before the indigenous vaccine’s introduction into the country’s UIP. Herd protection has been seen in other settings [37], and if high coverage has been achieved with other rotavirus vaccines, it may be difficult to demonstrate effectiveness of Rotavac®.”, Pg 16 lines 21, 22 & Pg 17 lines 1-3

Reviewers comment: Suppl Table 2 Project activities table1. Meaning of row 2 "source from which..." is unclear. Does this mean staff member? Also under table2, what aspect of form does 'legibility' refer to and Table 5. How is storage of samples assessed.

Changes made in the Supplementary Table which now reads Supplementary Table 3 in Pg 29-31. Changes made in Pg 29 line 10, Pg 30 line 2 and Pg 31 line 1 of 'main document marked copy'.

Reviewers comment: In summary indicators, as per above, can the study team reference past experience in getting >90% stool collection and >75% vaccination records.

In surveillance studies conducted in the past, we have obtained 85.7% of stools.

Ref: National Rotavirus Surveillance Network, Kumar CPG, Venkatasubramanian S, et al. Profile and Trends of Rotavirus Gastroenteritis in Under 5 children in India, 2012 - 2014, Preliminary Report of the Indian National Rotavirus Surveillance Network. Indian Pediatr 2016; 53:619–22.

In this study, which is now ongoing, we are able to obtain >90% of stool samples and >75% of vaccination cards from all the sites.

VERSION 2 – REVIEW

REVIEWER	Daniel Hungerford Institute of Infection and Global Health, University of Liverpool, UK
REVIEW RETURNED	11-Dec-2018

GENERAL COMMENTS	The authors have answered the majority of mine and my fellow reviewers comments, but please could they add the following:
---

	In the strengths and limitations bullet points - please state whether the authors consider the test-negative design a strength or limitation. As requested previously in the analysis plan please could they add how they intend to compare pre- and post vaccine rates of disease, i.e time-series or simple before and after comparison.
--	--

VERSION 2 – AUTHOR RESPONSE

Reviewer: 1

Reviewer Name: Daniel Hungerford

Institution and Country: Institute of Infection and Global Health, University of Liverpool, UK

Please state any competing interests or state 'None declared': None declared

The authors have answered the majority of mine and my fellow reviewers comments, but please could they add the following:

Comment 1: In the strengths and limitations bullet points - please state whether the authors consider the test-negative design a strength or limitation.

The strengths and limitations section in the main paper now reads as:

- First project in India to evaluate the impact and effectiveness of a newly introduced vaccine in children
- Use of test negative control design, a convenient and low cost technique used in effectiveness studies for several vaccines, including rotavirus vaccines
- Hospital based surveillance might not be representative of illness or vaccine coverage in the community

Page 11, lines 2-6 in the main document now reads as: “The test negative study design is a convenient and low cost method for estimation of effectiveness of a vaccine as the study can be facility- rather than community-based and since commercially available enzyme-linked immunosorbent assay (ELISA) kits have a sensitivity and specificity of over 99% [32], the probability of any bias is negligible [33]”.

Comment 2: As requested previously in the analysis plan please could they add how they intend to compare pre- and post vaccine rates of disease, i.e time-series or simple before and after comparison.

Author’s response: A simple before and after comparison will be done for comparing pre- and post-vaccine rates of rotavirus diarrhea.

Page 14, lines 5-10 in the main document now reads as: “Trends in all-cause diarrhea will be compared for pre- and post-vaccine introduction using available population demographics for five NRSN sites, Tirupati (SVMC); Bhubaneswar (Hitech hospital); Rohtak (PGIMS); Tanda (RPGMC) and Vellore (CMC) that have hospital based pre-vaccine data on AGE among under five children from 2012 onwards. For these 5 sites, a simple comparison of rotavirus diarrhea before and after the vaccine introduction will be estimated.”